# The Effects of Chest Wall Loading on Perceptions of Fatigue, Exercise Performance, Pulmonary Function, and Muscle Perfusion

**DOI:** 10.3390/sports8010003

**Published:** 2020-01-01

**Authors:** Gaia Giuriato, Anders Gundersen, Sarina Verma, Ethan Pelletier, Brock Bakewell, Stephen J. Ives

**Affiliations:** 1Health and Human Physiological Sciences Department, Skidmore College, Saratoga Springs, NY 12866, USA; gaia.giuriato@univr.it (G.G.); agunders@skidmore.edu (A.G.); sverma@skidmore.edu (S.V.); epelleti@skidmore.edu (E.P.); bbakewe@skidmore.edu (B.B.); 2Department of Neuroscience, Biomedicine and Movement Science, University of Verona, 37100 Verona, Italy

**Keywords:** chest wall restriction, work of breathing, load carriage, performance

## Abstract

**Background:** Load carriage (LC), which directly affects the chest wall and locomotor muscles, has been suggested to alter the ventilatory and circulatory responses to exercise, leading to increased respiratory muscle work and fatigue. However, studies exploring the impact of LC on locomotion increased internal work, complicating their interpretation. To overcome this issue, we sought to determine the effect of chest wall loading with restriction (CWL + R) on cycling performance, cardiopulmonary responses, microvascular responsiveness, and perceptions of fatigue. **Methods:** In a randomized crossover design, 23 young healthy males (22 ± 4 years) completed a 5 km cycling time trial (TT) in loaded (CWL + R; tightened vest with 10% body weight) and unloaded conditions. After baseline pulmonary function testing (PFT; forced expiratory volume in 1 s, FEV_1_; forced vital capacity, FVC), cardiopulmonary indices (HR, heart rate; O_2_ uptake, VO_2_; ventilation, V_E_; tidal volume, V_T_; and breathing frequency, B*_f_*), rating of perceived exertion (RPE), lactate (BLa), and microvascular responses (oxy-, deoxy-, total hemoglobin; and tissue saturation; StO_2_) of the vastus lateralis using near infrared spectroscopy were collected during the TT; and PFT was repeated post-exercise. **Results:** Pre-exercise, CWL + R reduced (*p* < 0.05) FVC (5.6 ± 0.8 versus 5.5 ± 0.7 L), FEV_1_ (4.8 ± 0.7 versus 4.7 ± 0.6 L), and FEV_1_/FVC (0.9 ± 0.1 versus 0.8 ± 0.1). CWL + R modified power output (PO) over time (interaction, *p* = 0.02), although the 5 km time (461 ± 24 versus 470 ± 27 s), V_T_ (3.0 ± 0.3 versus 2.8 ± 0.8 L), B*_f_*, V_E_, HR, VO_2_, microvascular and perceptual (visual analog scale, or VAS, and RPE) responses were unchanged (*p* > 0.05). CWL + R increased (*p* < 0.05) the average BLa (7.6 ± 2.6 versus 8.6 ± 3 mmol/L). **Conclusions:** Modest CWL + R negatively affects pre-exercise pulmonary function, modifies cycling power output over time, and increases lactate production during a 5 km cycling trial, although the cardiorespiratory, microvascular, and perceptual responses were unaffected.

## 1. Introduction

Load carriage (LC) can be defined as locomotion while bearing a mass upon the torso supported by shoulder straps and/or a hip belt (i.e., a backpack) [1]. LC is essential in certain recreational and occupational activities such as hiking, or military and emergency services [2,3]. This type of equipment varies from a total weight of a couple kilograms (kg) to an average of 22 kg for firefighting personal protective equipment and self-contained breathing apparatus, or more for military applications. This weight imposes an additional stress to cardiopulmonary and locomotor systems, which negatively impacts exercise tolerance, performance, and pulmonary function [4,5,6,7]. The placement of an extra load and/or restriction to the thoracic cavity causes a volume and movement limitation of the chest wall coupled with an increase in the work of breathing [6,8]. Thus, it is critical to understand how LC impacts basal physiology and the integrated response to exercise.

During exercise, inspiratory demands rise in concert with expiration, becoming an active process resulting in the increased recruitment of abdominal, scalene, and internal intercostal muscles, further elevating the need for O_2_ uptake and delivery. As ventilatory demand increases, the end-expiratory lung volume (EELV) decreases with a consequent increase in tidal volume (V_T_) and in breathing frequency (B*_f_*) [9,10,11]. With the increase of exercise intensity, end-inspiratory lung volume (EILV) increases to compensate the limited capacity of EELV to decrease, and to allow V_T_ to increase further [12]. In addition, as respiratory demand increases, changes in operational lung volume may occur to optimize the muscle length–tension relationship and avoid an expiratory flow limitation (EFL). This expiratory limitation occurs when the flow rate ceases to increase, with the increasing of expiratory effort (i.e., due to mechanical limitation by chest wall or airway obstruction) and the impairment of further increases in pulmonary ventilation (V_E_) during maximal performance [13]. CWL likely alters EILV, feasibly altering respiratory muscle length–tension relationships, resulting in suboptimal muscle function, greater metabolic cost, and conceivably ultimately fatigue.

Past studies demonstrated that carrying a 15–25 kg load in a backpack increased the electromyography signals of the external intercostals and the sternocleidomastoid, which suggested greater respiratory muscle work [8,14]. The obvious consequence of this increase in respiratory work is respiratory muscle fatigue [8], which may negatively affect the performance [15]. However, such studies used heavy absolute loading protocols (≥15 kg) and were conducted while walking or standing, which increases the internal work of the task and metabolic cost (e.g., VO_2_) onto itself, thereby increasing ventilatory demands. However, much less is known about the isolated impact of chest wall loading (CWL) on a task that is not susceptible to increased internal work with LC or CWL, such as cycling. While cycling could be a superior experimental paradigm, there is also potential practical application to those who are obese, carry hydration packs or backpacks/messenger bags during cycling, or police officers wearing tactical vests while on bicycle patrol.

Moreover, to investigate the specific effects of chest wall restriction without loading, Coast et al. found that as the magnitude of restriction increased, maximal oxygen consumption (VO_2max_), V_E_, time to achieve max, and V_T_ decreased significantly, while B*_f_* at max was maintained [16]. Further, previous research has shown that CWL + R effectively increases oxygen consumption by the external intercostal muscles [17] and the degree of expiratory flow limitation [18]. Indeed, Harms et al. [19] demonstrated that in the context of limitations to cardiac output, altered work of breathing compromises leg blood perfusion because of increased respiratory demand, and thus respiratory muscle blood flow, which in turn likely alters metabolism and reduces exercise performance. However, no study to date has looked at modest CWL with restriction (CWL + R) to determine what may be causing the possible alteration in performance associated with increased loading and restriction of the thorax during cycling, such as reduced muscle perfusion [15].

Therefore, the purpose of this study was to determine the effect of CWL with restriction (CWL + R) during a cycling task, in order to avoid a large increase of internal work, on the perception of fatigue, cardiorespiratory and metabolic responses, exercise performance, and locomotor muscle perfusion. It was hypothesized that commensurate with an exaggerated cardiorespiratory response, CWL + R would impart a greater metabolic demand, lactate concentration, impaired muscle perfusion, and increased perceptions of fatigue, leading to impaired exercise performance.

## 2. Methods

### 2.1. Participants and General Procedures

Twenty-three young healthy active males aged 18–25 years old were recruited by public advertisement and word of mouth from Skidmore College and the surrounding area. To recruit healthy participants, those with body mass index (BMI) ≥30 kg/m^2^, current or recent (<6 months) smokers, and those with any history of cardiovascular, pulmonary, renal, musculoskeletal, neural, or metabolic diseases were excluded. Health history was collected using questionnaires (American College of Sports Medicine Pre-Participation Screening) to assess for eligibility. To be fit, participants must have exercised at least five days per week. With this inclusion criteria, we could include athletes whose BMI was overestimated (>25 kg/m^2^) as a consequence of higher muscle mass. Participants were asked to arrive ≥2 h post prandial and hydrated, rested without recent strenuous exercise (within 24 h), and without consumption of alcohol or caffeine for 12 h prior. To achieve adequate hydration, they were suggested to ingest 500 mL 2–3 h prior and 250 mL 15 min prior to arrival. Participants were instructed to maintain a diet similar in macronutrient composition and caloric content for the duration of the study, and to avoid all dietary supplements (e.g., multivitamins, etc.). Studies were conducted in a thermoneutral (21 ± 1 °C, 29 ± 12% relative humidity), and normobaric (approximately 750 mmHg) environment. All participants provided written informed consent prior to participation. The Institutional Review Board of Skidmore College approved the protocol (IRB#1412-432) and is in accordance with the most recent revision of the Declaration of Helsinki (2013 version).

### 2.2. Experimental Design

The study was conducted using a counterbalanced, repeated measures crossover design to determine the effects of CWL + R on physiology and performance. Participants were randomly assigned to the order in which they completed the CWL + R and control trials. Testing for each participant required three visits (completed within two weeks, >48 h apart). During the familiarization trial, each participant’s body weight was measured; then, the load of the vest was determined by calculating 10% of the body weight of each participant, which was added to the vest and a slight chest restriction was applied, just below the point of discomfort. The weighted vest (VersaFit Vest, Power Systems, Knoxville, TN, USA) was made of inelastic material and positioned on each participant’s torso and tightly secured, just below point of discomfort, with Velcro straps to induce modest restriction; the additional 10% of individual body weight was achieved with sand-filled bags positioned, in a balanced manner, throughout the pouches on the vest. In addition, during the familiarization, participants performed a similar 5 km time trial (TT) to acquaint with the weighted vest, Monark 828E cycle ergometer, pulmonary function testing, mouthpiece for ventilatory measures, and scales used to assess perceptions of fatigue. Specifically, a visual analog scale (VAS) and RPE scales were explained and tested during this session. During the experimental trials, participants were only able to see the distance covered and were encouraged to exercise at a maximal or near maximal intensity. The 5 km TT was preferred to longer tests because of its feasibility, the time constraints of the participants, and was practical in nature, as these participants were not highly experienced cyclists. Moreover, this time trial has been previously used to assess aerobic endurance performance as well as predict maximal lactate steady state intensity and 40 km time trial performance.

### 2.3. Experimental Visits

Upon arrival, participants were fitted with a heart rate (HR) monitor (H7, Polar, NY, USA) and the weighted vest for the CWL + R trial. Then, they were fitted and seated to a cycle ergometer (828E, Monark) to start the testing procedures. Once seated, their maximal voluntary contraction for handgrip (MVC_HG_) was established using an analog handgrip dynamometer (Lafayette Instruments, Indiana) [20], which was recorded to estimate potential effects on central fatigue. If a reduction in MVC_HG_ was observed for a muscle not directly involved in the cycling 5 km TT, particularly in an upright cycling posture as employed on the Monark ergometer, this could be indicative of central fatigue and a reduced global motoneuronal output [21]. Then, pulmonary function testing was performed to measure the forced expiratory volume in one second (FEV_1_) and forced vital capacity (FVC) (microspirometer, CareFusion, Ireland, UK) [22]. A minimum of three tests were performed as long as they met the acceptability criteria put forth by the American Thoracic Society and European Respiratory Society [23], and the highest values were recorded. Then, participants were instrumented with a mouthpiece and two ways non-rebreathe valve (Hans Rudolph 2700), nose clip, and the expiratory port was coupled to a metabolic cart (Parvomedics TrueOne 2400, Sandy, Utah) [24] to determine O_2_ uptake (VO_2_), CO_2_ production (VCO_2_), ventilation (V_E_), tidal volume (V_T_), and breathing frequency (B*_f_*). The Parvomedics cart utilizes a paramagnetic O_2_ sensor (0.1% accuracy, 0–25% range), infrared CO_2_ sensor (0.1% accuracy, 0–15% range), and heated Rudolph pneumotach (±2% accuracy, 0–800 L/min range), which calibrated as per manufacturer suggested guidelines. Finally, we employed near infrared spectroscopy (NIRS) of the vastus lateralis muscle (Oxiplex TS, ISS Champaign, IL, USA) to determine the potential effects of CWL + R on muscle microvascular responses (oxy- (OxyHb), deoxy- (DeoxHb), total hemoglobin (Hbtot), and tissue saturation (StO_2_)) as surrogates for blood flow (Nioka et al. 2006; Bopp et al. 2014; Jones et al. 2016). Prior to use, the probe was calibrated using a block with known absorption and scattering coefficients, and the exact placement on the muscle was marked with a permanent marker to maintain the consistency of probe placement between conditions. Participants were also shaved at the NIRS site, and the skin was cleaned with alcohol prep pads.

Then, participants quietly rested for 5 min while baseline ventilatory measurements (VO_2_, VCO_2_, V_E_, V_T_, B*_f_*) were recorded. In addition, resting perceptions of fatigue were measured via visual analog scale (VAS, 0 “no fatigue” to 100 “severe fatigue”) [25] and three separate ratings of perceived exertion (RPE) scales (from 1 to 10) [26]. The three perceptual scales consisted of total body RPE (RPE, “overall, how hard are you working?”), leg RPE (RPE_leg_, “how hard do you feel like your legs are working?”), and dyspnea (RPE_dyspnea_, “how hard do you feel like your breathing is?”). After this period, capillary blood lactate was obtained (Lactate Pro, Arkray, Minneapolis, MN, USA) from the middle finger of the non-dominant hand, although due to logistics, blood lactate was only obtained in 12 subjects. Then, participants were allowed a two-minute warm-up period at a self-selected pace and resistance, which was recorded and replicated exactly for each participant in their next trial. In the final 15 s of the warm up, the resistance was adjusted to the testing level of 2.5 kilopounds (kp), and upon instruction, the participant began the 5 km TT as fast as possible, giving a maximal effort. Thus, participants modulated power output (PO) via self-selecting cadence throughout the 5 km TT. With the use of RPM recordings, PO was calculated as PO (watts) = ((2.5kp × y RPM × 6m) ÷ 6). The 2.5 kp load was chosen, supposing a range of revolutions per minute (RPM) between 50 and 120 provided a wattage range of 125 to 300 during the 5 km trial, which, based upon prior work, facilitates the range of power outputs to be expected in the active college age population (Vidal et al. 2017). Previous work in our laboratory demonstrated that an adequate reliability of the 5 km TT is a 3.3% coefficient of variation and intraclass correlation coefficient = 0.92 (n = 15, unpublished prior observations). Verbal encouragement was offered during each trial in a consistent manner both within and between participants. During the 5 km TT, the three RPE scales, HR, and blood lactate were measured every minute, revolutions per minute (RPM) were measured every 30 s, and pulmonary measurements (VO_2_, V_E_, etc.) were measured every 15 s. The near infrared spectroscopy (NIRS) signal was obtained continuously. Immediately upon completion of the 5 km time trial, MVC_HG_, pulmonary function, and perceptual measures of fatigue were repeated. Afterwards, the vest was removed, and participants cooled down for 5 min. Participants reported back to the lab to complete their second experimental trial under the other condition to which they had completed their first trial no less than 48 h later.

### 2.4. Data Analysis

Statistical comparisons were performed with the use of the commercially available software (SPSS v. 22.0, IBM Inc., Armonk, NY, USA). Paired samples t-tests were used to identify potential changes at rest in response to CWL + R and to compare performance (5 km time, average, minimal, and peak power output), fatigue (VAS and MVC_HG_), and pulmonary function at rest, and in response to exercise. Effect size was determined using Cohen’s D (0.2 = small effect, 0.5 = medium effect, and 0.8 = large effect). Two-way repeated measures of analysis of variance (ANOVA) were used to determine if there was an interaction between condition (control versus CWL + R) and time, as well as to determine if the main effects existed for either condition or time. As per tradition, these results are presented as interaction finding first, followed by the main effect of condition and time. As time to trial completion varied between participants, the minimal time for which all participants had complete data (6 min, the fastest time) and the individual final time were entered into the statistical model. Further, the data analyzed as such are graphed accordingly, with data expressed over time to the last interval (fastest time), and each participant’s data upon completion of the 5 km TT. The level of significance was established at *p* < 0.05. All data were expressed as means ± standard deviation (SD), unless specified otherwise, where means are ± standard error of the mean (SEM).

## 3. Results

### 3.1. Participants Characteristics

The present investigation was performed on 23 healthy young subjects (height: 180 ± 3 cm; weight: 81.8 ± 4 kg; BMI: 25 ± 1 kg/m^2^) that performed both trials (Table 1).

### 3.2. Pulmonary Function

As compared to the control trial, pre-exercise pulmonary function was significantly reduced with CWL + R; as CWL + R reduced FVC (5.6 ± 0.8 versus 5.5 ± 0.7, *p* = 0.01), FEV_1_ (4.8 ± 0.7 versus 4.7 ± 0.6 L, *p* < 0.001), and FEV_1_/FVC (0.9 ± 0.1 versus 0.8 ± 0.1, *p* = 0.014), control versus CWL + R, respectively. Post-exercise, there was no significant difference in FEV_1_ (5.2 ± 0.8 versus 5.1 ± 0.7 L, *p* = 0.595), FVC (5.5 ± 0.8 versus 5.5 ± 0.8 L, *p* = 0.47), or FEV_1_/FVC ratio (0.9 ± 0.1 versus 0.9 ± 0.1, *p* = 0.72) between conditions (control versus CWL + R, respectively).

### 3.3. Performance

There was no statistically significant difference in 5 km time between the control and CWL + R condition (461 ± 59 versus 470 ± 66 s, *p* = 0.12). This time difference was approximately 9 s between trials, which corresponds to an approximately 2% increase in 5 km time in the CWL + R condition. During the 5 km time trial, there was a significant (*p* = 0.02) interaction between condition and time in PO, where CWL + R seemed to modify PO over time, particularly during the initial stages of exercise (Figure 1). PO at the onset of exercise was significantly higher in the control condition versus the CWL + R condition (245 ± 36 versus 231 ± 38 Watts, *p* = 0.004), which was more moderately paced. However, the peak PO observed during the 5 km TT was not statistically different between conditions (273 ± 34 versus 281 ± 36 Watts, *p* = 0.088, control versus CWL + R, respectively). In terms of cycling efficiency, the CWL + R condition (15.0 ± 0.3 mL/Watt) was not different from the control (15.3 ± 0.4 mL/Watt) in terms of interaction between condition and time, nor an effect of condition or time (all, *p* > 0.05).

### 3.4. Perceptions of Fatigue

There was no significant difference in pre-exercise MVC_HG_ (56 ± 7 versus 57 ± 9 kg, *p* = 0.231). Further, as expected, there was no significant difference between the pre-exercise perception of fatigue between conditions (9 ± 11 versus 13 ±13 mm VAS, *p* = 0.150, control versus CWL + R, respectively). Post-exercise, there was no significant difference in post-exercise MVC_HG_ (56 ± 7 versus 58 ± 8 kg, *p* = 0.578) or perception of fatigue (75 ± 17 versus 78 ± 14, *p* = 0.342) between conditions, control versus CWL + R, respectively. Expectedly, the perception of fatigue was greater post-exercise (*p* < 0.001), although the handgrip remained unchanged (*p* > 0.05). During exercise, there were no significant interactions between condition and time or a main effect of condition (CWL + R versus control) in the overall rating of perceived exertion (RPE, *p* = 0.821), leg RPE (RPE_leg_, *p* = 0.279), or dyspnea rating (*p* = 0.554). Expectedly, a significant time effect (*p* < 0.001) was observed for overall RPE (RPE), RPE_leg_, and dyspnea, where all perceptual measures of overall fatigue, leg specific fatigue, or breathing effort increased over the course of the 5 km time trial.

### 3.5. Cardiorespiratory and Metabolic Responses to Exercise

Baseline HR was not different between conditions (80 ± 13 versus 81 ± 10 bpm, *p* = 0.687). During exercise, there was no significant interaction between condition and time for HR (*p* = 0.389) or a main effect of condition (*p* = 0.682, Figure 2). Expectedly, there was a significant (*p* < 0.001) effect of time where, not surprisingly, HR increased throughout the 5 km TT in both conditions (Figure 2). However, the end exercise hear trates were not different between conditions (184 ± 11 versus 183 ± 18 bpm, Figure 2).

There were no significant interactions between condition (control versus CWL + R) and time for V_T_, V_E_, VO_2_, VCO_2_, B*_f_*, VE/VO_2_, VE/VCO_2_, or RER (all *p* > 0.05, Figure 3). Further, there were no significant effects of condition on V_T_ (3.0 ± 0.3 versus 2.8 ± 0.8 L, *p* = 0.131), V_E_ (101.6 ± 25.7 versus 96.7 ± 23.7 L/min, *p* = 0.187), VO_2_ (43.7 ± 6.9 versus 42.2 ± 6.6 mL/kg/min, *p* = 0.085), VCO_2_ (47.8 ± 9.3 versus 47.5 ± 8.9 mL/kg/min, *p* = 0.385), B*_f_* (37 ± 4 versus 37 ± 5 breaths/min, *p* = 0.428), VE/VO_2_ (28.4 ± 0.9 versus 28.5 ± 0.8), VE/VCO_2_ (25.8 ± 0.9 versus 25.8 ± 0.7), or RER (1.1 ± 0.1 versus 1.1 ± 0.1, *p* = 0.982), control versus CWL + R, respectively (all, *p* > 0.05, Figure 3). However, there was a significant main effect of time (all, *p* < 0.001) where cardiorespiratory responses expectedly increased throughout the 5 km TT in both conditions. However, there was a significant interaction of the condition (control versus CWL + R) and time (*p* < 0.001) for lactate concentration (Figure 4), where lactate (BLa) values were higher at the end of the 5 km TT in CWL + R condition (10.7 ± 2.8 versus 12.2 ± 2.4 mmol/L). Although, the area under the curve for BLa was not significantly different between conditions (0.40 ± 0.02 versus 0.40 ± 0.01).

### 3.6. Near Infrared Spectroscopy

There were no significant interactions between condition (Control versus CWL + R) and time for Hbtot, HbO, DeoxHb, or StO_2_ (all, *p* > 0.05, Figure 5). There were also no significant effects of condition on Hbtot (95.1 ± 13.9 versus 82.6 ± 7.3 µM, *p* = 0.296), HbO (53.0 ± 10.5 versus 46.8 ± 5.9 µM, *p* = 0.268), DeoxHb (41.7 ± 3.9 versus 35.7 ± 2.3 µM, *p* = 0.840), and StO_2_ (%: 54.3 ± 2.2 versus 56.1 ± 1.2, *p* = 0.482) (Figure 5). As expected, there was a significant effect of time for Hbtot, HbO, DeoxHb, and StO_2_ (Figure 5) in response to the 5 km TT (all, *p* < 0.001).

## 4. Discussion

The purpose of this study was to investigate the influence of chest wall loading with modest restriction (CWL + R) on cycling exercise performance, pulmonary function, ventilation, muscle perfusion, and fatigue. A principal finding from the present study was that CWL + R significantly reduced pre-exercise pulmonary function, lowering FEV_1_, FVC, and FEV_1_/FVC by 160 mL, 100 mL, and 1%, respectively. During the 5 km TT, CWL + R modified cycling PO over time, resulting in an on average slower 5 km time (approximately 9 s) that was not statistically significant. Interestingly, perceptual indices of fatigue were the same during and post-exercise, despite the altered pacing strategy in PO invoked by the CWL + R. Metabolically, while oxygen consumption was unchanged, the lactate production was significantly higher. Microvascular responses were not changed by the CWL + R condition, as were ventilation, VO_2_, efficiency, and ventilatory equivalents for VO_2_ and VCO_2_. Modest levels of chest wall loading (10% body weight) and restriction (100–160 mL reduction in FVC and FEV_1_) resulted in minimal reductions in non-weight bearing exercise performance (approximately 2%), although the physiological (e.g., increased lactate) and conscious or subconscious (e.g., pacing) strategies to achieve the performance likely differed.

### 4.1. Effect of CWL + R on Pulmonary Function

During unloaded eupneic breathing, increasing tidal volume (V_T_) occurs at the expense of end-inspiratory lung volume (EILV) and end-expiratory lung volume (EELV). With a mass load on the chest (i.e., a backpack, military equipment, etc.), this mechanism is altered: left uncompensated, EILV is increased, reducing V_T_ due to the weight on the chest wall [27]. In an attempt to maintain V_E_, either breathing frequency must increase or V_T_ is compensated through increased primary and secondary respiratory muscle activity, which translates in an increase of the energy cost of breathing [6,28]. Such a reduced efficiency in the respiratory muscles, coupled with the likely disruption of muscle length tension relationships, reduced elastic recoil associated with impaired EILV, or simple mechanical perturbations are possible mechanisms in the observed reduction pre-exercise FVC and FEV_1_ [16]. Irrespective of the mechanism, in the current study with CWL + R, there was a consistent, albeit modest, reduction in canonical indicators of pulmonary function (Figure 5).

In support of the current findings, different studies determined that CWL and a constant-pressure chest wall-restrictive device reduced baseline FVC and FEV_1_ [16,29]. However, usually the ratio of FEV_1_ to FVC is maintained, which opposed the current study’s findings where the FEV_1_/FVC ratio was reduced. Interestingly, post-exercise, no such difference existed, which was possibly due to the relatively limited load and/or restriction. The present study demonstrated that relatively modest CWL + R produced significant reductions in pre-exercise pulmonary function, including both FVC and FEV_1_. These findings suggest that those with pre-existing pulmonary limitations (e.g., obesity, chronic obstructive pulmonary disorder, etc.) consider alternatives to load carriage other than on the thorax, and that such alteration could possibly translate into a negative impact the normal ventilatory response exercise, particularly in vulnerable populations.

### 4.2. Effect of CWL+R on the Cardiorespiratory Responses

Prior investigations into the effects of CWL, such as those of Coast et al., found that restrictive CWL during submaximal cycling significantly decreased V_T_ with a compensation in B*_f_*, resulting in similar VO_2_ and V_E_ [16]. In the same study, during maximal exercise, VO_2_, V_E_, and V_T_ were significantly reduced, while B*_f_* was maintained [16]. However, Gonzalez et al. found a significant increase in the oxygen cost associated with external chest wall restriction, which was proportional to the severity of chest wall restriction [17]. In the current study, despite the participants performing a maximal effort test, as the RER was ≥1.10 and the peak ventilatory equivalents ≥30, the model of loading and restriction that was applied did not challenge V_T_ enough to induce a detectable change in cardiorespiratory parameters, as compared to the restrictive challenge employed by Coast et al. [16] or Gonzalez et al. [17]. It is important to note that most studies have employed LC while walking, and the study on cycling listed above was conducted purely with an aggressive model of chest wall restriction without a load mass. Moreover, the unchanged ventilatory equivalents (VE/VO_2_ and VE/VCO_2_) between conditions reflect a relatively normal exchange efficiency despite CWL + R. Collectively, using modest restriction and loading, we found no difference in cardiorespiratory responses to high intensity exercise, despite a pacing strategy, suggesting that modest load carriage (10% or less of body weight), exerts modest or null effects on the cardiorespiratory responses.

### 4.3. Effect of CWL + R on Metabolic and Hemodynamic Matching

Previous research [17] has shown that CWL + R effectively increases oxygen consumption by the external intercostal muscles. Indeed, Harms et al. [19] demonstrated that exercise at maximal oxygen consumption compromises leg blood flow because of increased respiratory demand. In agreement, Vogiatzis et al. [30] established that the circulatory system in cyclists is unable to meet the demands of both active skeletal muscle tissue and the respiratory intercostal muscle during heavy exercise. However, in the current study, we did not observe a reduction in muscle perfusion, as assessed by total hemoglobin (Figure 5) at the vastus lateralis, although it may be possible that blood was diverted away from other muscle active during cycling (e.g., hips flexors, other muscles of the quadriceps) and toward the respiratory muscles to overcome the hypothesized increased resistance/work of breathing associated with CWL + R. Alternatively, although end 5 km heart rates were approximately 90% of age-predicted max, it is possible that there was enough cardiac reserve to provide any possible increased cardiac output demands. Interestingly, CWL + R significantly increased lactate concentrations compared to control (Figure 4). However, we did not observe a reduction in pulmonary VO_2_ in the CWL + R trial, which could be interpreted to reflect the shift in blood flow away from the large muscle mass engaged in cycling to the respiratory muscles, which was affectionately termed the “steal effect”, leading to reduced oxygen availability and increased lactate. However, measures of respiratory muscle perfusion and peripheral direct Fick VO_2_ lactate efflux measures would be needed to definitively support this hypothesis in this model. A novel aspect of this study was the use of cycling to investigate the impact of CWL + R on perceptions of fatigue, avoiding an increase of internal work due to an additional load in walking/running tasks.

### 4.4. Effect of CWL + R on Perception of Fatigue and Performance

As with the development of limb locomotor muscle fatigue [31], the genesis of respiratory muscle fatigue is also a complex process that can originate both centrally or peripherally. Indeed, while previous work has indicated reductions in perfusion and VO_2_ [19] or time to exhaustion [16], whether these phenomena translate to reduced performance is relatively unknown, as is the effect on fatigue. Thus, in part, to evaluate the potential decrease in performance potentially due to fatigue, perceptual measures of total, dyspnea, and leg exertion were obtained, as well as a visual analog scale for fatigue. In contrast to our initial hypotheses, our perceptual measures of fatigue were not different between trials, although given that PO was modified over time, this is perhaps suggestive of a greater response relative to the PO.

Although initial PO was lower in the weighted CWL + R condition, the average PO results were similar to those of the controls (Figure 1). This was reflected with an on average minimal increased time to complete the 5 km trial, although it was not significant, and was likely the result of a pacing strategy. Hussain et al. [32] supports this altered exercise capacity with chest wall restriction and found that during short-term, constant load, heavy exercise on a cycle ergometer test at 80% of maximum PO to exhaustion, interference with ribcage expansion reduced endurance time. In relative agreement, although the model employed was different, Coast et al. [16] found that during a graded maximal cycling test to exhaustion, the time to maximal exercise was significantly decreased in the trial with restrictive chest wall loading. However, that we found no signficant effect on the 5 km time suggests that our level of CWL + R was insufficient to elicit a robust reduction in performance, or perhaps the exercise task was of insufficent duration.

Interestingly, not all previous studies have recorded or reported a perception of effort or fatigue [6,16,33,34,35], but in the current study, we find that RPE, RPE_leg_, and RPE_dyspnea_ were relatively unaffected by CWL + R. Tomczak et al. [28] reported a significantly increased RPE_dyspnea_ in the restricted condition throughout exercise, but they found no difference in RPE_leg_ between the control and the chest wall restriction condition. Thus, perhaps in the current experimental paradigm, the CWL + R was not severe enough, or the exercise was not long enough in duration to induce a greater rating of effort for breathing, for the legs, or the overall rating of perceived exertion. Finally, the model used in this study employed both CWL (10% body weight) and restriction (albeit minimally, as the average tidal volume was only approximately 126 mL lower during the 5 km TT). Thus, we cannot determine whether the observed effects are attributable to loading or restriction, and future studies should consider employing a vest-only trial to partition out the potential contribution of each.

### 4.5. Limitations

This study is not without limitations. Specifically, while the vest was tightened to just below the point of discomfort, we had no way to quantify the magnitude of restriction, aside from the changes in pulmonary function, and as these effects were modest, there is some question as to whether they are meaningful. Further, ad hoc, it was determined to make some measurements based upon time and not distance, potentially limiting interpretation. Although we sought to use multiple methods to estimate fatigue, they are all indirect or perceptual measures, and future studies could consider using more quantitative techniques such as twitch interpolation to have more quantifiable measures of fatigue. Moreover, the choice to perform a time trial test, without a fixed pace and power output, where participants were able to manipulate cadence and thus power output, limits the ability of the current study to assess the three phases, or components, of VO_2_. Future studies should explore the possible effects of CWL + R in longer distances (20 or 40 km TT) with greater magnitude (>10% body weight) or should use constant load exercise to investigate the different domains of VO_2_ to better understand the potential influences of CWL + R on physiology during cycling.

## 5. Conclusions

The current study is the first to investigate the combination of modest CWL with restriction during exercise in a non-weight bearing model of cycling performance to attempt to parse out the effects of altered total work versus the potential influence of CWL. CWL + R negatively affected pre-exercise pulmonary function and increased lactate concentrations during the 5 km time trial. In contrast to what was initially hypothesized, CWL + R modified the power output over time, possibly eliciting a pacing strategy, but did not alter perceptions of fatigue, cardiorespiratory and metabolic responses, exercise performance, or perfusion of the active muscle. Overall, a decreased performance was not observed, which was probably due to the inadequate load and restriction used, although the physiological and conscious or subconscious (e.g., pacing) strategies to achieve the performance likely differ.

## Figures and Tables

**Figure 1 sports-08-00003-f001:**
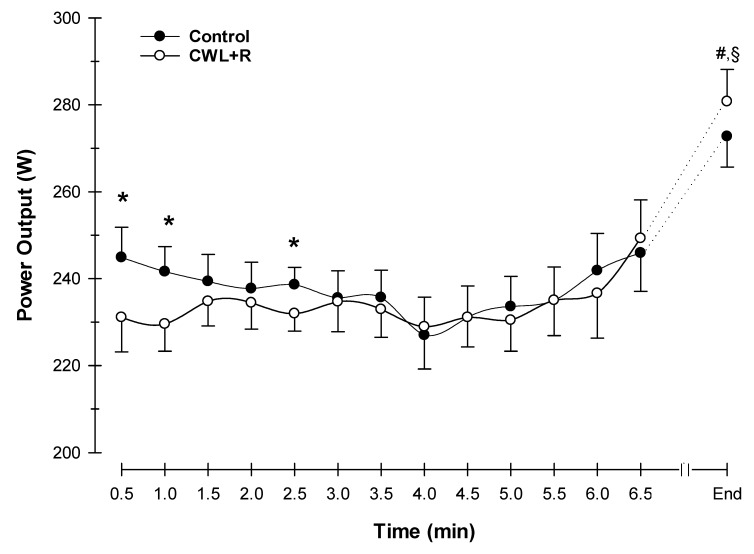
Effect of CWL + R on cycling power output during the 5 km TT (N = 23). Due to variation in the individual time required for completion of the task, data were analyzed and presented to the point where all the participants presented data (minute 6.5) + the final individual data point (end of 5 km TT). Values are means ± SEM. Condition x time effect: *p* = 0.02; Condition effect: *p* = 0.366; Time effect: *p* < 0.001. * *p* < 0.05 CWL + R vs. control; # *p* < 0.05 difference from baseline for CWL+R; ^§^
*p* < 0.05 difference from baseline for control.

**Figure 2 sports-08-00003-f002:**
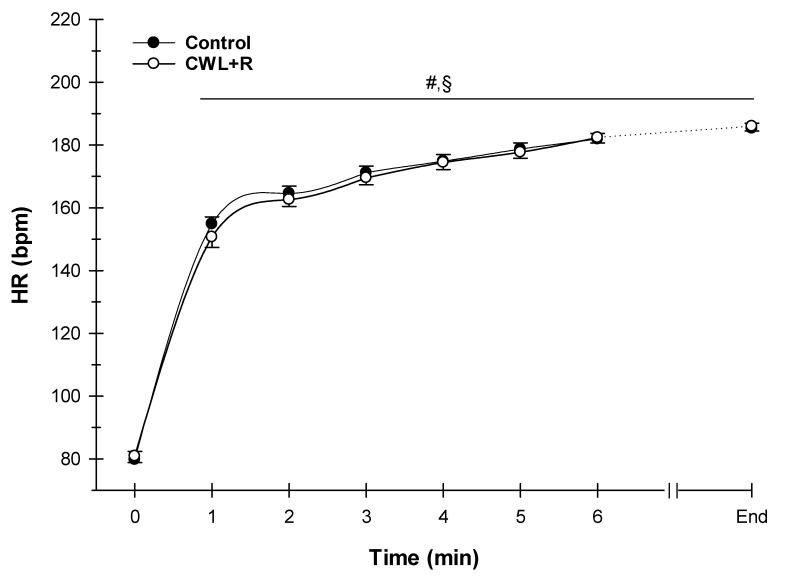
Effect of CWL + R on heart rate (HR) during the 5 km TT (n = 23). Due to the variation in individual time required for completion of the task, data were analyzed and presented to the point where all the participants presented data (minute 6) + the final individual data point (end of 5 km TT). Time × condition effect: *p* = 0.389; Condition effect: *p* = 0.682; Time effect: *p* < 0.001. Values are means ± SEM. ^#^
*p* < 0.05 difference from time 0 for CWL + R; ^§^
*p* < 0.05 difference from time 0 for control.

**Figure 3 sports-08-00003-f003:**
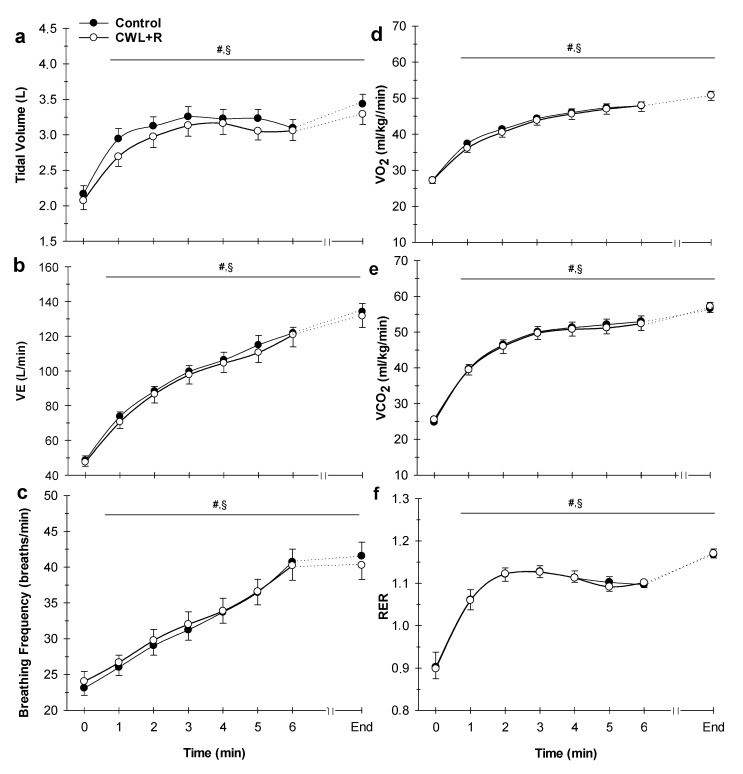
Effect of CWL + R on ventilatory and metabolic parameters during the 5 km TT (n = 23). (**a**) Tidal volume, (**b**) ventilation, (**c**) breathing frequency, (**d**) oxygen consumption (VO_2_), (**e**) carbon dioxide production (VCO_2_), and (**f**) respiratory exchange ratio (RER). Due to the variation in individual time required for completion of the task, data were analyzed and presented to the point where all the participants presented data (minute 6) + the final individual data point (end of 5 km TT). All, Time effect: *p* < 0.001. Values are means ± SEM. ^#^
*p* < 0.05 difference from time 0 for CWL + R; ^§^
*p* < 0.05 difference from time 0 for control.

**Figure 4 sports-08-00003-f004:**
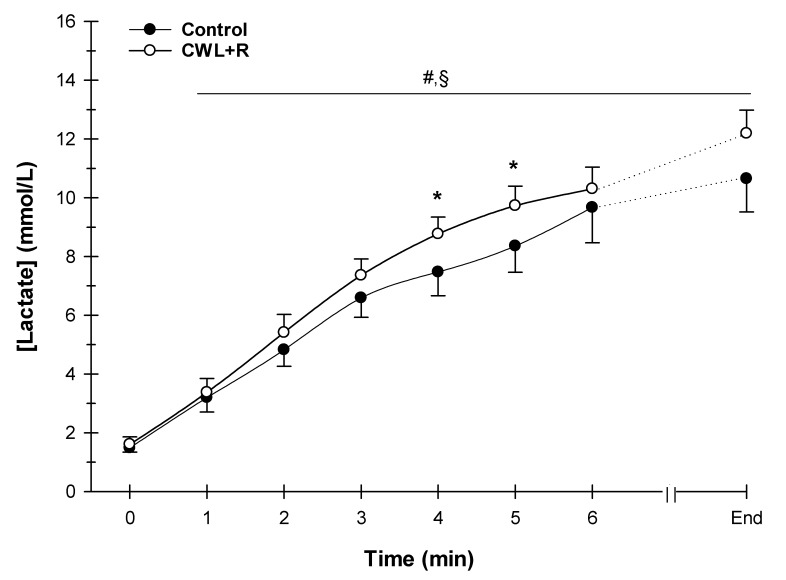
Effect of CWL + R on capillary blood lactate during the 5 km TT (n = 12). Due to the variation in individual time required for completion of the task, data were analyzed and presented to the point where all the participants presented data (minute 6) + the final individual data point (end of 5 km TT). Values are means ± SEM. Interaction of condition by time *p* < 0.001. * *p* < 0.05 CWL + R vs. control; ^#^
*p* < 0.05 difference from time 0 for CWL + R; ^§^
*p* < 0.05 difference from time 0 for Control.

**Figure 5 sports-08-00003-f005:**
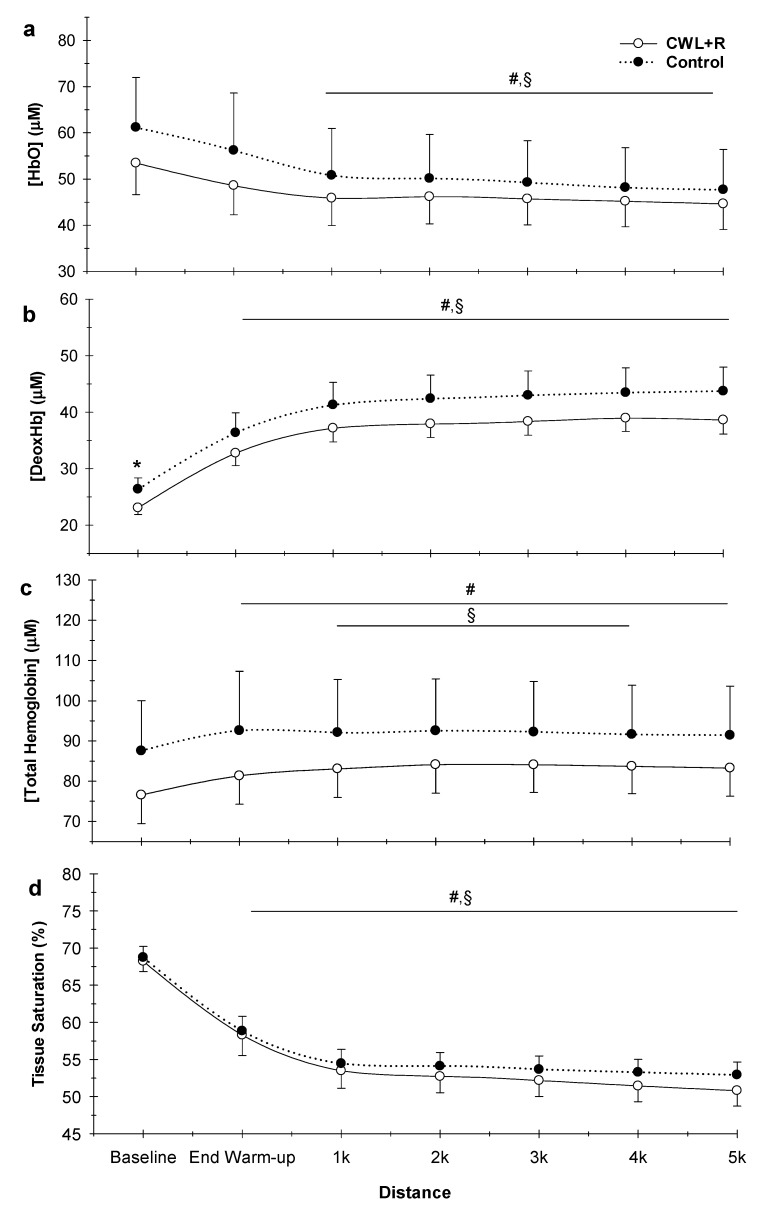
Effect of CWL + R on the microcirculation during the 5 km TT (n = 12). (**a**) Oxyhemoglobin, (**b**) deoxyhemoglobin, (**c**) total hemoglobin, and (**d**) tissue oxygen saturation (StO_2_). All, condition x time effect: *p* > 0.05; Condition effect: *p* > 0.05; Time effect: *p* < 0.001. Values are means ± SEM. * *p* < 0.05 CWL + R vs. control; ^#^
*p* < 0.05 difference from baseline for CWL + R; ^§^
*p* < 0.05 difference from baseline for control.

**Table 1 sports-08-00003-t001:** Respiratory and perceptual indicators of fatigue at baseline and the peak response during the 5 km time trial (TT) in chest wall loading with restriction (CWL+R) and control conditions.

Variable	Baseline		Cohen’s d	Peak		Cohen’s d
	CWL + R	Control		CWL + R	Control	
**V_T_, L/min**	2.1 ± 0.6	2.2 ± 0.6	0.18	3.3 ± 0.6 †	3.4 ± 0.7 †	0.06
**V_E_, L/min**	47.5 ± 10.8	48.5 ± 10.8	0.1	133.0 ± 29.0 †	135.9 ± 31.9 †	0.14
**VO_2_, L/min**	2.2 ± 0.4	2.2 ± 0.4	0.02	4.1 ± 0.6 †	4.1 ± 0.6 †	0.23
**VO_2_, mL/kg/min**	27.2 ± 4.0	27.3 ± 3.7	0.02	50.8 ± 6.4 †	50.8 ± 7.2 †	0.23
**VCO_2_, L/min**	2.1 ± 0.4	2.1 ± 0.7	0.14	4.6 ± 0.6 †	4.6 ± 0.8 †	0.13
**VCO_2_, mL/kg/min**	25.5 ± 4.8	24.5 ± 6.2	0.14	57.2 ± 7.8 †	57.4 ± 7.6 †	0.13
**RER**	0.88 ± 0.10	0.87 ± 0.10	0.03	1.17 ± 0.05 †	1.17 ± 0.05 †	0.08
**V_E_/VO_2_**	21.7 ± 3.2	21.8 ± 3.0	0.01	31.3 ± 8.3 †	33.1 ± 4.8 †	0.30
**V_E_/VCO_2_**	23.6 ± 4.7	24.4 ± 4.2	0.21	29.9 ± 5.4 †	30.3 ± 4.6 †	0.09
**B_f_, breaths/min**	18 ± 1	18 ± 1	0.1	37 ± 3 †	37 ± 4 †	0.09
**Fatigue_VAS_ (mm)**	13 ± 10	9 ± 11	0.41	78 ± 14†	75 ± 17 †	0.18
**RPE_leg_**	2.3 ± 0.9	2.7 ± 1.0	0.41	6.9 ± 1.5 †	7.3 ± 1.6 †	0.15
**RPE_dyspnea_**	1.6 ± 0.8	1.6 ± 0.8	0.00	5.8 ± 1.7 †	6.0 ± 1.7 †	0.12
**RPE_tot_**	2.0 ± 0.7	2.2 ± 0.9	0.31	6.9 ± 1.2 †	6.8 ± 1.4 †	0.07

**Notes:** Data are presented as mean ± standard deviation (n = 23). †: *p* < 0.05 baseline vs. peak within condition. V_T_ = tidal volume; V_E_ = ventilation; VO_2_ = oxygen consumption; VCO_2_ = carbon dioxide production; RER = respiratory exchange ratio; V_E_/VO_2_ = ventilatory equivalent for oxygen; V_E_/VCO_2_ = ventilatory equivalent for carbon dioxide; B_f_ = breathing frequency; Fatigue_VAS_ = visual analogue scale; RPE = rate of perceived exertion.

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
