# Peer review of "The Effects of Chest Wall Loading on Perceptions of Fatigue, Exercise Performance, Pulmonary Function, and Muscle Perfusion"

_sports, 2020, doi:10.3390/sports8010003_

Round 1
Reviewer 1 Report
Comments
Overall this is a nice idea for a study and highly novel, at least to my knowledge. There seem to be some controls missing in parts e.g. what are the differences between groups before adding the CWL? Could these just be inherent differences between groups and nothing to do with the CWL? The 5km time trial needs justifying as it's incredibly short - the shortest time trials are 20km and Olympic distance is 40km. Perhaps you would have found greater differences over longer more applicable distances?
I think the manuscript may be suitable for acceptance following addition of further details and information requested below.
Intro
Apologies if I have missed this in the intro or if there is literature I am not familiar with, but how would increased load carriage lead to a reduction in muscle perfusion? If it increases work would this not increase cardiac output and thereby increase muscle perfusion?
Methods
You seem to have excluded participants with a BMI over 30. Most subjects with a BMI over 25 would be classed as overweight (unhealthy) so what was the rationale for excluding only obese individuals?
Line 93 - exercised 5d/wk. any further details? These sound like very highly trained individuals and if this is the case - why the BMI of over 30 as an exclusion and not 25?
Line 93 - further details on subjects to allow replication e.g. VO2max of participants?
Line 102 - please specify which revision of Declaration of Helsinki rather than just say most recent
What was the rationale for a 5km time trial? This is very short - olympic TT is 40km for example? Just a sentence or two to explain the rationale would be helpful please
Line 177 - why were t tests completed as opposed to one-way ANOVA? Does this method control for the inflation of type I error?
Results
Line 200 - p = 0.12 could this be because the study was underpowered for this measure?
Figure 1 - why is there no baseline at 0 min? Were the groups assessed before the addition of the vest to see if any differences existed between groups independent of the chest-wall loading?
Figure 2 - please make all y axis the same size for ease of comparison
Line 442 - what is the full P value please where you have 0.00?
Figure 5 - nice figure. Might I suggest you look at area under the curve in both conditions. I think this may be more sensitive to what your data are showing as it seems clear lactate is higher from 2 mins onwards and might give a clearer overall picture of what is happening
Figure 6 - is baseline before or after chest wall loading? Do you have data to show there aren't differences between the groups independent of the chest wall loading?
Discussion
Line 246 - if you're interested in cycling performance why do a 5km time trial? I'm not sure a 5k cycling time trial exists within international cycling? Something like 20km is normally the minimum distance. Please explain the rationale for this.
Line 257 - Just a thought on your lactate. How tight was the chest wall loading? Is there any possibility this could have compressed the chest cavity sufficiently to put pressure on the liver and influence the cori cycle? Speculation on my part... just a thought
Line 306-307 - have you any data on time trial time? Was this slower in the CWL condition? apologies if I've missed these data
Line 322 -as above, were they slower over the TT indicating fatigue?
Line 330-332 - so they weren't significantly slower. How meaningful was the CWL if this didn't reduce performance? Can we claim this led to fatigue etc if not sig different?
Line 366 - you mention a pacing strategy. Do you have data to back this up? Please discuss these data if so
References
These need to be updated. There are 20 references and not one from the last three years. This doesn't give the impression of pushing the field along if all references to previous work are 10-20 years old. There is room for seminal work but these must be balanced with how this advancing the field (currently)
Author Response
We want to thank the reviewers for the effort in the revision process. We have responded to the comments in a point-by-point fashion below, highlighting where changes were made as appropriate.
REVIEWER #1
Overall this is a nice idea for a study and highly novel, at least to my knowledge. There seem to be some controls missing in parts e.g. what are the differences between groups before adding the CWL? Could these just be inherent differences between groups and nothing to do with the CWL?
RESPONSE: Thank you for this comment. As this was a crossover design study the two trials had the same, and same number, of participants. Probably, it is a visual problem because when the data between the two conditions are really similar, the white symbol for the control condition is hidden behind the black signal for the CWL+R condition. The participants of the two groups were the same, and the test were performed in two different days in order to let the participants recover properly (line 107: Testing for each participant required three visits (completed within 2 weeks, >48 hours apart)). We double checked and we confirm that the baseline values before adding the CWL+R are the same between groups.
The 5km time trial needs justifying as it's incredibly short - the shortest time trials are 20km and Olympic distance is 40km. Perhaps you would have found greater differences over longer more applicable distances?
RESPONSE: Correct, 5km TT is short by road cycling standards. We opted for this choice because this time trial has been previously used to assess aerobic endurance performance, to predict maximal lactate steady state intensity and 40Km time trial performance (https://onlinelibrary.wiley.com/doi/full/10.1111/j.1600-0838.2004.00424.x). Further, the individuals who participated were trained but not highly experienced cyclists who would be familiar with the 20 and 40km TT, thus we would have expected more variability in the 40Km time trial which lasts nearly one hour for cycling athletes, whereas the 5Km TT lasts from 7 to 12 minutes. The 5km has also been used in previous physiological research https://www.ncbi.nlm.nih.gov/pmc/articles/PMC2670040/ . The reviewer raises an interesting point if longer distances might yield a greater effect, we have added this to the limitations section of the discussion.
Intro
Apologies if I have missed this in the intro or if there is literature I am not familiar with, but how would increased load carriage lead to a reduction in muscle perfusion? If it increases work would this not increase cardiac output and thereby increase muscle perfusion?
RESPONSE: We apologize this was unclear, we have since revised this statement. The reviewer is correct, if exercise and cardiac output are submaximal, cardiac output could be increased, though in maximal effort scenarios, as in the current study, cardiac output is already nearly maximized, thus a redistribution of cardiac output is the alternative strategy. In agreement, Harms et al. showed that exercise at maximal oxygen consumption increased work of breathing, such as with load carriage perhaps, compromises leg blood perfusion because of increased oxygen and blood flow demands to respiratory muscles, which has been referred to as the “steal effect”. Moreover, Vogiatzis et al. established that the circulatory system in trained athletes is unable to meet the demands of both active skeletal muscle tissue and the respiratory intercostal muscle during heavy exercise. We thought that adding a load/resistance on the thorax, in order to increase the respiratory work, this phenomenon would have occurred.
Methods:
You seem to have excluded participants with a BMI over 30. Most subjects with a BMI over 25 would be classed as overweight (unhealthy) so what was the rationale for excluding only obese individuals? Line 93 - exercised 5d/wk. any further details? These sound like very highly trained individuals and if this is the case - why the BMI of over 30 as an exclusion and not 25?
RESPONSE: The decision for this exclusion criteria was preceded by the rationale of this study. Load Carriage is a condition of work for e.g. military and firefighters. However, a lot of healthy and fit firefighters have a 25<BMI<30. From Kuehl et al. (Body Mass Index is a Predictor of Fire Fighter Injury and Worker Compensation Claims): “Twenty-five percent of firefighters were in the normal weight category (BMI 18.5 ≤ 24.9). Fifty-six percent of the firefighters were in the overweight category (BMI 25≤ 29.9). Nineteen percent of the firefighters were in the obese weight category (BMI > 30).” Moreover, BMI is a measure that tend to overestimate the values in tall athletes or with an important muscle mass. To better clarify this issue, a description has been added in the text: “To be fit, participants must have exercised at least five days per week. With this inclusion criteria, we could include athletes whose BMI was overestimated (>25 kg/m2) as a consequence of high muscle mass.”
Line 93 - further details on subjects to allow replication e.g. VO2max of participants?
RESPONSE: Unfortunately, participants didn’t perform an incremental test to assess VO2max. We can only show the peak Vo2 for the trial, that has been inserted in Table 1. Some previous work suggests that the VO2 during a 5km represents ~85% of VO2max, which would be an average of 58 ml/kg/min, though this is purely an estimation only. Thank you for this suggestion.
Line 102 - please specify which revision of Declaration of Helsinki rather than just say most recent
RESPONSE: We added in the text the version of the Declaration of Helsinki: “…is in accordance with the most recent revision of the Declaration of Helsinki (2013 version).”
What was the rationale for a 5km time trial? This is very short - olympic TT is 40km for example? Just a sentence or two to explain the rationale would be helpful please
RESPONSE: An explanation has been added in the text (lines 121-124): “The 5km TT was preferred to longer tests because of its feasibility, time constrains of the participants, and minor incidence of injuries. Moreover, this time trial has been previously used to assess aerobic endurance performance, to predict maximal lactate steady state intensity and 40Km time trial performance.”
Line 177 - why were t tests completed as opposed to one-way ANOVA? Does this method control for the inflation of type I error?
RESPONSE: Paired samples t-tests were used as the levels of the independent variable was 2 (control vs. CWL+R), only leaving 1 degree of freedom for simple comparisons, such as 5km time. A one way would only be appropriate when the levels of the IV is more than 2 and the df more than 1. However, as originally included in the manuscript, we did use two-way repeated measures ANOVA for comparing the 2 conditions over multiple time points (e.g. 5km). Neither approach controls for type I error, though in such a study we are more likely to encounter a type II error, as in the case of the effect on CWL+R on 5km time with a p value of 0.12.
Results:
Line 200 - p = 0.12 could this be because the study was underpowered for this measure?
RESPONSE: It is possible, as there are two main contributors, effect size and number of participants. Probably, a greater weight would have elicited a greater effect size and/or measuring more people, might have shown significance. With a robust 23 participants on this measure I suspect the issue might be more of the former, that is a small effect size (Cohens d = 0.2).
Figure 1 - why is there no baseline at 0 min? Were the groups assessed before the addition of the vest to see if any differences existed between groups independent of the chest-wall loading?
RESPONSE: Thank you for this comment. We decide to avoid the results of Power Output from the baseline because the participants were not cycling and it would have been 0 Watt. Moreover, the pace and workload were self-selected each time for the individuals at both time points.
Figure 2 - please make all y axis the same size for ease of comparison
RESPONSE: The figure 2 has been removed, and the related data has been added to Table 1.
Line 442 - what is the full P value please where you have 0.00?
RESPONSE: The p values in the figures that were written P=0.00 has been changed with P<0.001, as in the text. Thank you
Figure 5 - nice figure. Might I suggest you look at area under the curve in both conditions. I think this may be more sensitive to what your data are showing as it seems clear lactate is higher from 2 mins onwards and might give a clearer overall picture of what is happening
RESPONSE: Thank you, and we appreciate the suggestion, the area under the curve data has been added in the text. However, they are not statistically different.
Figure 6 - is baseline before or after chest wall loading? Do you have data to show there aren't differences between the groups independent of the chest wall loading?
RESPONSE: Baseline was assessed with participants on the bike, after 15minutes that CWL+R was on.
Discussion:
Line 246 - if you're interested in cycling performance why do a 5km time trial? I'm not sure a 5k cycling time trial exists within international cycling? Something like 20km is normally the minimum distance. Please explain the rationale for this.
RESPONSE: Cycling performance was preferred to a running task to advert possible changes in physiological responses due to different running strategies between the two trials. The 5k TT has been chosen because participants were not specifically cyclists and because this test has been seen in the previous literature to predict tests like the 40km TT.
Line 257 - Just a thought on your lactate. How tight was the chest wall loading? Is there any possibility this could have compressed the chest cavity sufficiently to put pressure on the liver and influence the cori cycle? Speculation on my part... just a thought
RESPONSE: An interesting thought. Unfortunately, we had no way to quantify the magnitude of restriction, aside from the changes in pulmonary function. We asked the participants to tight the vest just below the point of discomfort. Though the restriction was probably not enough to influence liver blood flow or in a meaningful way the cori cycle.
Line 306-307 - have you any data on time trial time? Was this slower in the CWL condition? apologies if I've missed these data
RESPONSE: No, time trials were not statistically different. The CWL+R was on average 9 seconds slower than the control. The data are better shown in the results section: “There was no statistically significant difference in the final 5km time between the control and CWL+R condition (461 ± CI: [437, 485] vs 470 ± CI: [443, 497] seconds, p=0.12, Cohen’s d=0.16). This time difference of ~9 seconds, corresponds to a 2% increase in 5km time in the CWL+R condition.”
Line 322 -as above, were they slower over the TT indicating fatigue?
RESPONSE: No, they were statistically not slower, and in accordance the scales for the perception of fatigue were not different.
Line 330-332 - so they weren't significantly slower. How meaningful was the CWL if this didn't reduce performance? Can we claim this led to fatigue etc if not sig different?
RESPONSE: Probably, CWL+R that we used was not enough to induce a significant effect. As we are aware of this limitation, in lines 338-340 it is explained with “Though given we found no significant effect on 5km time suggests our level of CWL+R was insufficient to elicit a robust reduction in performance or perhaps the exercise task was of insufficient duration.” and with the section of the Limitations “This study is not without limitations. Specifically, while the vest was tightened to just below the point of discomfort, we had no way to quantify the magnitude of restriction, aside from the changes in pulmonary function, and as these effects were modest there is some question as to whether they are meaningful. … Though, we sought to use multiple methods to estimate fatigue, they are all indirect or perceptual measures, and future studies could consider using more quantitative techniques such as twitch interpolation to have more quantifiable measures of fatigue.”
Line 366 - you mention a pacing strategy. Do you have data to back this up? Please discuss these data if so
RESPONSE: No, we don’t have data to support this statement, for this reason we only assume that was the result of a pacing strategy. Lines 330-333: “Although initial PO was lower in the weighted CWL+R condition, suggestive of a pacing strategy, the average PO was similar to the control condition (Fig. 1). This was reflected with an, on average, minimal increased time to complete the 5 km trial, although not significant, and was likely the result of a pacing strategy.” We have also added a qualifier to line 363 to connotate this is conjecture.
References:
These need to be updated. There are 20 references and not one from the last three years. This doesn't give the impression of pushing the field along if all references to previous work are 10-20 years old. There is room for seminal work but these must be balanced with how this advancing the field (currently)
RESPONSE: Thank you, we have added references published in the current or last couple of years.
Reviewer 2 Report
Dear authors,
This is an original works where authors have written a good contextualization and a discussion. Nevertheless, authors haven´t included some necessaries variables like VO2 slow component, the phases of VO2 response to exercise, RER or VE/VO2 and VE/VCO2. Additionally, I think that it´s necessary to include some tables for a better understanding of the results. Moreover, I want to realize the next specifics commentaries:
Line 49: “…frequency (Bf) (Younes and Kivinen 1984; Sharratt et al. 1987; Henke et al. 1988)” the references must be adapted to the journal´s guidelines. Lines 92-93: “participants must have exercised at least five days per week” wasn´t it questioned about the volume and intensity of the exercise? In the section 2.3 the references must be revised attending to the journal´s guidelines. For a completed study of the ventilatory response it´s necessary to include the next variables: RER, VE/VO2 and VE/VCO2. Additionaly, I think that it´s necessary to analyze the three phases of the VO2 response in an exercise of this characteristics, the slow component of VO2 and mechanical efficiency. Additionally, I propose to include an analysis for each km (not for minutes). Authors must change some figures and transform it in tables. The results section must be rewritten including tables and figures. It´s impossible to understand the results like authors have presented the results.
Author Response
We want to thank the reviewers for the effort in the revision process. We have responded to the comments in a point-by-point fashion below, highlighting where changes were made as appropriate.
This is an original works where authors have written a good contextualization and a discussion. Nevertheless, authors haven´t included some necessaries variables like VO2 slow component, the phases of VO2 response to exercise, RER or VE/VO2 and VE/VCO2. Additionally, I think that it´s necessary to include some tables for a better understanding of the results. Moreover, I want to realize the next specifics commentaries:
RESPONSE: Thank you for the suggestions. We included in the text and tables the RER, VE/VO2 and VE/VCO2.
Line 49: “…frequency (Bf) (Younes and Kivinen 1984; Sharratt et al. 1987; Henke et al. 1988)” the references must be adapted to the journal´s guidelines.
RESPONSE: The citation has been updated. Thank you!
Lines 92-93: “participants must have exercised at least five days per week” wasn´t it questioned about the volume and intensity of the exercise?
RESPONSE: The five days per week statement refers solely to a minimal inclusion criterion to consider the participants fit; the actual intensity or volume of the exercise was not quantified. Though most they were all collegiate athletes.
In the section 2.3 the references must be revised attending to the journal´s guidelines.
RESPONSE: A revision of all the references has been done. Thank you!
For a completed study of the ventilatory response it´s necessary to include the next variables: RER, VE/VO2 and VE/VCO2.
RESPONSE: We included in the text and tables the RER, VE/VO2 and VE/VCO2.
Additionaly, I think that it´s necessary to analyze the three phases of the VO2 response in an exercise of this characteristics, the slow component of VO2 and mechanical efficiency.
RESPONSE: Thank you for the suggestion. However, since workload varied over time and with CWL, an analysis of the slow component would be imprecise and the interpretation muddied.
Additionally, I propose to include an analysis for each km (not for minutes).
RESPONSE: Unfortunately, we cannot match the time with the km. We have to say that since the time between the two trial was similar, the trial can be considered similarly distance matched.
Authors must change some figures and transform it in tables.
RESPONSE: Thank you for the suggestion. We added Table 1 to better understand the ventilatory patterns (included RER, VE/VO2 and VE/VCO2) and the RPE. As a consequence, the RPE figure was deleted.
The results section must be rewritten including tables and figures. It´s impossible to understand the results like authors have presented the results.
RESPONSE: Yes, we completely agree. Results are now written with Mean ± Standard Deviation.
Round 2
Reviewer 1 Report
Thank you for addressing my comments in the initial report.
Author Response
We are appreciative of the reviewers comments and efforts in improving the current manuscript.
Reviewer 2 Report
Dear authors,
I want to grant you the inclusion of the variable RER in a table and a figure and the inclusion of VE/VO2 and VE/VCO2 in a table. Nevertheless, VE/VO2 and VE/VCO2 hasn´t been included like abbreviations. Additionally, these variables hasn´t been included in the discussion. Authors have corrected some references and they have clarified a little more the 2.1. section.
Despite the changes commented before, authors hasn´t considered any of my suggestion. The more important limitation and the poorest section of the paper is the results section. This section is horrible. So, in Table 1 must be included the p-value of the t-test for baseline and peak values. Additionally, I suggest to include ES. Additionally, the figures are wrong. So, it´s necessary to include the statistical differences of the Post-Hoc. Also, the results must be rewritten (the text of this section is too confusing.
The result section need an important revision. Additionally, it´s necessary to include the discussion of RER; VE/VO2 and VE/VCO2 variables. Additionally, I believe that the inclusion of the phases of VO2 is very suitable, but mechanical efficiency is necessary.
If authors don´t consider all my comments, this paper couldn´t be accepted for publishing.
Author Response
I want to grant you the inclusion of the variable RER in a table and a figure and the inclusion of VE/VO2 and VE/VCO2 in a table. Nevertheless, VE/VO2 and VE/VCO2 hasn´t been included like abbreviations. Additionally, these variables hasn´t been included in the discussion. Authors have corrected some references and they have clarified a little more the 2.1. section.
RESPONSE: We appreciate the recognition our of initial, now recognized as insufficient revisions, and we apologize for the oversight. We have since made significant changes in response to the comments here and below. We completed the abbreviation section with VE/VO2 and VE/VCO2. Moreover, VE/VO2, VE/VCO2 and RER have been added to the discussion (lines 294-302). Please see specific responses below, changes to the manuscript have been highlighted in red text.
Despite the changes commented before, authors hasn´t considered any of my suggestion. The more important limitation and the poorest section of the paper is the results section. This section is horrible.
RESPONSE: We wish to thank reviewer 2 for their constructive feedback and sincerely apologize if we did not respond fully in the last round of revisions. We believe we have since adequately addressed the concerns of the reviewer, herein and within the manuscript where appropriate (lines 198-253). Further, to provide better clarity for the reader, we also revised the statistical analysis to indicate our intention with the results section (lines 184-191). Thank you for your time and efforts in improving this manuscript, we feel it is now much improved.
So, in Table 1 must be included the p-value of the t-test for baseline and peak values. Additionally, I suggest to include ES.
RESPONSE: Table 1 has been significantly revamped. It is now divided in Baseline results, comparing CWL+R v and Control, followed by the Peak values observed during the 5Km TT between conditions. In addition, Cohen’s d values have been in Table 1 for comparison between conditions, as well as markers (†) for significant p-values of the t-test for baseline vs peak values. Thank you for this suggestion.
Additionally, the figures are wrong. So, it´s necessary to include the statistical differences of the Post-Hoc. Also, the results must be rewritten (the text of this section is too confusing.
RESPONSE: Regrettably, the figures are a representation of the data we found. Inferring the issue the reviewer raised was inclusion of the post-hoc results, we have since added this information to the figures. Further, this comment prompted us to reflect upon the figures and have since visually revised the figures to more clearly reflect the fact that the data are, as mentioned in the methods and figure titles, expressed discontinuously in Figures 1-4.
The result section need an important revision. Additionally, it´s necessary to include the discussion of RER; VE/VO2 and VE/VCO2 variables. Additionally, I believe that the inclusion of the phases of VO2 is very suitable, but mechanical efficiency is necessary.
RESPONSE: Thank for these suggestions. As mentioned previous we have significantly revised the results section and believe it is now clearer. VE/VO2, VE/VCO2 and RER have been added to the discussion (lines 296-304). In addition, cycling efficiency has been added as well in the text (lines 213-215). However, regarding the phases of VO2, we feel it is important to clarify a critical point, that is to understand properly the different phases of VO2 it’s necessary that the workrate (wattage) is constant (Please see review by Jones and Colleagues in MSSE, 43, 11, 2011). In fact, work rate varied significantly over time in the present study precluding an effectual measurement of the components of VO2. Quantifying the slow component for example, would likely be capturing an altered power output, such as at the end the TT, and not the actual phase III of VO2, and thus such a quantification would be erroneous. Thus, we cannot in good faith complete this request and respectfully disagree it is appropriate in the current study. However, as we do believe that such an exploration of CWL+R on the components of VO2 would be interesting and novel, we now mention this in the discussion section (lines 369-374), as this may be an interesting follow up study. Thank you.
If authors don´t consider all my comments, this paper couldn´t be accepted for publishing.
RESPONSE: We appreciate the comments above and have made significant changes to the manuscript. Thank you again for your efforts in improving the current manuscript, they are appreciated.
Round 3
Reviewer 2 Report
Dear authors,
I want to grant you the enhancement in the manuscript. Now, the manuscript have improved significantly. The inclusion of VE/VO2 and VE/VCO2 and the efficiency are very important. The results are more understandable now. I think that the only change necessary is to specify in the statistical analysis the inclusion of effect size (ES) and the implication of each range of values. After this minor inclusion, I will recommend to accept this manuscript for publishing.
Author Response
I want to grant you the enhancement in the manuscript. Now, the manuscript have improved significantly. The inclusion of VE/VO2 and VE/VCO2 and the efficiency are very important. The results are more understandable now. I think that the only change necessary is to specify in the statistical analysis the inclusion of effect size (ES) and the implication of each range of values. After this minor inclusion, I will recommend to accept this manuscript for publishing.
RESPONSE: Thank you, we have since included the ES in the methods with interpretations.